# Relationship between Body Composition and Cardiac Autonomic Regulation in a Large Population of Italian Olympic Athletes

**DOI:** 10.3390/jpm12091508

**Published:** 2022-09-14

**Authors:** Daniela Lucini, Antonio Spataro, Luca Giovanelli, Mara Malacarne, Raffaella Spada, Gianfranco Parati, Nadia Solaro, Massimo Pagani

**Affiliations:** 1BIOMETRA Department, University of Milan, 20129 Milan, Italy; 2Exercise Medicine Unit, Istituto Auxologico Italiano, IRCCS, 20135 Milan, Italy; 3Sports Medicine Institute CONI, largo G. Onesti 1, 00197 Rome, Italy; 4Department of Endocrine and Metabolic Medicine, Istituto Auxologico Italiano, IRCCS, 20149 Milan, Italy; 5Department of Medicine and Surgery, University of Milano-Bicocca, 20126 Milan, Italy; 6Department of Cardiology, Istituto Auxologico Italiano, IRCCS, 20149 Milan, Italy; 7Department of Statistics and Quantitative Methods, University of Milano-Bicocca, 20126 Milan, Italy

**Keywords:** autonomic nervous system, heart rate variability, sport, exercise training, elite athletes, performance, body composition, multivariate statistics

## Abstract

Athletic performance is determined by many factors, such as cardiorespiratory fitness, muscular strength and psychological features, which all interact simultaneously. The large Italian National Olympic Committee database of Olympic athletes offers a unique healthy population to verify the strength of the interplay among a number of major elements of training, including autonomic nervous system (ANS) modulation, biochemical indicators and body composition, in a system medicine approach. This observational, retrospective study involved 583 individuals. As part of the yearly precompetitive examination, cardiac autonomic (heart rate variability), psychological, physical (cycloergometer stress test), biochemical and body composition (BOD POD) evaluations were performed. In subsequent analysis, we first considered the relationship between body composition and single individual variables in a simple correlation matrix, including a multitude of variables; then, Exploratory Factor Analysis (EFA) restricted the information to six latent domains, each combining congruent information in relation to body composition. Finally, we employed a multiple quantile regression model to evaluate possible relationships between ANSIs (index capable of synthetizing ANS regulation) and the latent domains indicated by EFA reflecting body composition. We observed a clear relationship between ANS and body mass composition parameters, as indicated by both bivariate correlations and the quantile regression result of ANSIs versus the latent domain aggregating mainly body composition data expressed in % (*p* = 0.002). In conclusion, these results suggest that specific training may elicit parallel adaptation of ANS control and body composition. The analysis of Olympic athletes’ data allowed us to obtain a better understanding of the complex, multidimensional factors involved in determining sport performance. The latter appears to be determined by the simultaneous interaction not only of cardiorespiratory fitness, muscular strength and psychological features, but also of ANS cardiovascular modulation and body composition.

## 1. Introduction

Olympic athletes represent a unique group among top sport performers, as they work hard for many consecutive years not just to be allowed to participate in a competition, but to gain the first position on the podium.

We previously figured that in addition to optimal physical training, accurately quantified through the assessment of aerobic fitness and maximal oxygen uptake [1], other elements, such as psychological features and cardiovascular autonomic regulation may play a key role in determining a winning phenotype [2]. The assessment of body weight and body composition, moreover, offers a different, complementary view on the factors involved in the determination of the athletic performance, according to the specific specialty and to its weight sensitivity [3]. In fact, the description of the degree of overall performance in athletes should also consider that well-known determinants, such as fitness and muscular strength [4], may simultaneously interact with other factors, suggesting that a system approach might allow a more complete understanding [5]. Furthermore, athletic performance is highly dependent on neuropsychological skills that are trainable [6], interact with specific hormones, such as leptin [7], and are sensitive to individual stress levels [8]. Finally, as an integrated function, exercise is based on the dynamical interaction of a hierarchical neural network [9], whereby the “central command” [10] co-activates both somatic and autonomic circuits in order to maintain homeostasis [11] in the context of changing demands from the periphery, which are specific for every sport category [12]. Moreover, a few decades ago, it became clear that a major part of this complex activity is orchestrated by the autonomic nervous system, which could be seen as a key component of exercise performance [13] and of cardiac adaptation to training [14].

In the last decades, analysis of the beat-by-beat variability (V) of the heart period (or rather its inverse: heart rate, HR) gained wide acceptance as a means to obtain proxies of autonomic regulation [15]. Heart rate variability (HRV) appears particularly useful in sports as a means to follow various steps of training [16], as does using miniaturized portable instruments [17]. Over the last few years, the practical use of HRV in sport [16] has focused on time-domain variables (essentially considering various indices of HRV, such as standard deviation), while frequency-domain techniques, such as spectral analysis, where considered to be still of research nature [18], and in spite of the interest elicited by the potential capacity to distinguish between the vagal and sympathetic drives [19] and their balance [20], are particularly useful in sport [21]. This approach may also account for the multivariate relationship between cardiovascular autonomic regulation and other systems’ components, such as body composition (e.g., muscle mass) [22], inflammation [23], obesity [24] or stress [25], all orchestrated by a central organization [26]. In this context, in addition to the well-known importance of the sympathetic nervous system [27], recent experimental studies have provided convincing evidence of a direct influence of vagal activity on weight control [28], thus furthering the model of sympathetic and parasympathetic antagonistic interaction [26] beyond cardiovascular regulation. Investigators’ interest towards a more extensive assessment of the relationship among biochemical indicators, body composition [3] and autonomic regulation is, however, limited [29] by technical barriers: the lack of standardization, possible intrusiveness, variable accuracy of the evaluation of body composition, complexity of the assessment of autonomic regulation and cost. Considering the autonomic nervous system, the complexity of analytical techniques and difficulty of interpretation, in particular if mechanistically considering one index at a time, may act as barriers to the more extensive use of HRV in practice, particularly during dynamic conditions, such as exercise [30]. In order to overcome some of these barriers, we introduced an integrated, unitary Autonomic Nervous System Index for sports (ANSIs) [31], free of age and gender bias, using a ranking evaluation (from 0–100, higher is better) based on a large benchmark population. This index may facilitate the practical comprehension of data derived from autonomic nervous system study, permitting an easy use in clinical and/or sport settings; for instance, comparing ANSIs values obtained in the same individual over time in different training statuses, or defining different autonomic profiles characterizing different sport specialties [31]. Considering body composition, in recent years the introduction of a simple, inexpensive and friendly technique based on air plethysmography (BOD POD) [32,33] permits a reasonably accurate [34] assessment of essential elements of body composition (fat mass and fat free mass, both expressed in absolute physical units, kg, and nominal % value) to be obtained [35], employing specific equations according to a given population [3]. In this paper we used the large Italian database of elite Olympic athletes [36] to verify the strength of the interplay among the autonomic nervous system, biochemical indicators, body mass and body composition in a system medicine approach [5].

## 2. Materials and Methods

This observational, retrospective metabolic sub-study involved data from 583 (353 males and 230 females; median age ± MAD (median absolute deviation): 24 ± 4 and 23 ± 4 years, respectively; Mann–Whitney test: *p* = 0.842) elite athletes who took part in the selection procedure for the 2016 Rio Olympic Games for the National Italian Olympic Committee (CONI) [36]. The study protocol was guided by the STROBE statement [37]. Abbreviations and definitions of variables are presented in Table 1; general anthropometric characteristics of this population are presented in Table 2.

As part of the preparticipation screening, all athletes underwent a complete history and clinical evaluation, which included (i) a supine and upright indirect autonomic assessment by way of autoregressive (AR) spectral analysis of RR Variability [20], (ii) a standardized symptom-limited incremental bicycle ECG stress test, (iii) an assessment of body composition, (iv) subjective stress symptom profiling, determined with a self-report questionnaire that we introduced during the preparation for the Athens Olympic Games [8], and (v) routine blood test analysis. This procedure took place at the Institute of Medicine and Sports Science, CONI, Rome. All individuals whose data were utilized for this study were cleared for active sports participation and were thus free from evidence of diseases or disturbances that could interfere with results.

This study followed the principles of Declaration of Helsinki and Title 45, US Code of Federal Regulations, Part 46, Protection of Human Subjects, Revised 13 November 2001, effective 13 December 2001, and was cleared by local Institutional Science Committee (Istituto Medicina e Scienza dello Sport, Roma, Italy) and by Ethics Committee of University of Milan (report dated 23 September 2019). All subjects gave their written informed consent.

### 2.1. Protocol

Subjects arrived at the clinic between 9.00 and 12.00 a.m., at least 2 h after eating a light breakfast. After the initial clinical assessment, inclusive of a subjective stress evaluation and routine biochemical analysis (a summary list is reported in Table 1), they underwent the following tests: body composition, autonomic evaluation, incremental stepwise bicycle exercise test.

#### 2.1.1. Body Composition

Body composition was determined with air plethysmography, using BOD POD (Life Measurements Inc., Concord, CA, USA). To maintain calibration within predetermined accuracy, athletes enter the system wearing only a bathing suit and a cap. In stable resting conditions, two measurements are obtained and their average is utilized to provide body volume (in L), body mass (in kg) and derive % fat mass (FM) and % fat free mass (FFM) with specific equations.

#### 2.1.2. Autonomic Evaluation and Exercise Stress Test

All subjects were requested to avoid caffeinated beverages since waking and to keep physical training in the preceding 24 h within intermediate levels. After a preliminary 10-min rest period in the supine position for stabilization and sensor placement, an ECG, arterial pressure waveforms (Finometer, TNO, NL) and respiratory activity were continuously recorded over a nominal 5-min period of supine rest (rest) followed by 5 min standing up (stand). The ECG (CM5) and the respiratory signal were obtained with a two-way radiotelemetry system (Marazza, Monza Italy). Data were acquired with a PC at 250 samples/channel/s using a parabolic interpolation to improve R peak detection accuracy. We used [20] an autoregressive algorithm to automatically compute power and frequency of spectral components in the bandwidth of interest, discarding components of <5% power that are treated as noise [38]. The software tool was set so as to consider components with a center frequency of 0.03–0.14 Hz as low frequency, and components within the range 0.15–0.35 Hz as high frequency, recalling that “the HF component is synchronous with the respiration” [20], using a high coherence between RR variability and respiration as a confirmation. Autoregressive model generally considered parameters from 10 to 12. Recordings of subjects with low-frequency breathing were discarded to avoid entrainment and biased increased LF power. Additionally, recordings presenting arrhythmias were not considered.

Moreover, athletes underwent an incremental stepwise bicycle exercise stress test that was calibrated to last about 8–12 min, and was always performed as a last item of the medical evaluation, in order to avoid the autonomic bias occurring after exercise. As previously described, we employed two custom software tools: Heart Scope [38] to obtain a series of indices of autonomic regulation through autoregressive spectral analysis of RR variability under supine rest and during standing up [20] (see Table 1); and Dynascope [39] to obtain acceleration and deceleration capacity [40] during exercise and recovery. Symbolic dynamics of HRV were also assessed [41] and presented only to allow comparison with other studies. Considering the sympathetic–parasympathetic antagonistic dynamics within the “unbroken” unitary purpose of neural visceral regulation [26], we introduced a unitary Autonomic Nervous System Index for sports (ANSIs) [31]. ANSIs is built as a radar plot synthesis of a set of variables likely to individually contribute to overall cardiac autonomic regulation in sports and thus treated as proxies of autonomic indices of RR interval regulation [24]. ANSIs integrates the functional information related to cardiac autonomic modulation at rest (RR, RR variance, HF nu), under gravitational stimulation (change from rest to stand, ∆ LF nu) and during exercise (bradyslope and deceleration capacity) [42]. Thus, all critical elements contributing to determine the cardiac autonomic modulation profile during exercise [43] are integrated into ANSIs.

#### 2.1.3. Subjective Somatic Stress Symptoms

As in a prior study [8], all subjects completed a self-administered questionnaire with Likert scales about: (i) the appraisal of overall stress and (ii) fatigue perception from 0 (“no perception”) to 10 (“highest perception”); (iii). The subjective stress-related somatic symptoms questionnaire (4S-Q), inquires about 18 somatic symptoms accounting for the majority of somatic complaints. For scoring purposes, responses were coded from 0 (“no feeling”) to 10 (“a strong feeling”); thus, the total score ranged from 0 to 180 [8].

### 2.2. Statistics

Statistical analyses were performed on the variables listed in Table 1 according to a non-parametric and exploratory approach. Descriptive data were reported as median ± MAD, and the significance of differences between the two gender groups was tested through the non-parametric Mann–Whitney (MW) test (2-tailed). Nominal significance level was set at 0.05. The subsequent analysis was performed in three steps.

(1) Spearman correlation matrices of a representative subset of the study variables were computed separately for males and females. We employed a color-coding schema to highlight the strength of significance between individual indices and facilitate appreciation of clustered correlations. The correlation coefficient (r) for all the distinct pairwise comparisons within the two groups is reported in Figure 1a,b.

(2) In addition, we applied Exploratory Factor Analysis (EFA) (with the principal component extraction method and varimax rotation) [44] to synthesize the amount of information carried by the multitude of relevant experimental variables (23 in all plus gender) that were selected from Table 1 considering their informative clinical content. As previously described [45], EFA produced a smaller number of uncorrelated common latent factors, corresponding to latent domains, from the entire set of the considered variables without assuming any a priori assumptions on the data. We kept in analysis the first common factors such that they reproduced together at least 70% of the total variance, as indicated by the Variance Accounted For measure (VAF, in %), and, singularly, not less than 5% of the total variance. These common factors were then interpreted using the factor loadings (i.e., correlation coefficients between factors and variables) greater than or equal to 0.4 in absolute value. If a single variable appeared to correlate highly with more than a single factor (latent domain), it was attributed to the latent domain with which it had the highest loading. The latent domains thus obtained may suggest hypotheses about the nature of the modeling data structure and attribute a physiological meaning to groups of variables.

(3) Finally, we employed a multiple quantile regression model [46] to evaluate possible relationships between ANSIs (a proxy of autonomic nervous system control), considered as the dependent variable with the median as the quantile to be predicted, and the latent domains indicated by EFA, regarded as the model independent variables. In particular, we provided approximate estimates of the latent domain effect sizes based on the standardized beta coefficients and the Pseudo R^2^ index.

Computations were performed with commercial statistical packages (SPSS v 28, IBM, Armonk, NY, USA; Excel, Microsoft 365, Microsoft, Redmond, DC, USA).

## 3. Results

Table 1 reports the abbreviations and definitions of variables employed in this study.

Table 2 reports summary anthropometric data for the entire study population. Notably, age was essentially the same in the two gender groups (*p* = 0.842), while, as expected, weight, height, arterial pressure, waist circumference and body mass index were significantly greater in males (*p* < 0.001).

Table 3 also reports summary data regarding body composition, as determined by air plethysmography (BOD POD, see Methods). As expected, all indices differed significantly (*p* < 0.001) in the two gender groups. In particular, males showed a smaller FM% and higher FFM% and a directionally similar result regards absolute (Kg) fat mass and fat free mass. Body mass, body volume and BSA (body surface area) were, as customary, greater in the male group.

Considering major metabolic indicators (Table 4), total and HDL cholesterol were lower and glucose was higher in males, although remaining always within normal ranges. Regarding subjective indices of stress (Table 4), both self-reported indices of stress and fatigue were similar in the two groups; conversely, the 4SQ (subjective stress-related somatic symptoms) score was lower in males. These data were not processed further. They are only presented to demonstrate a low involvement of the stress mechanisms in this healthy subpopulation of elite athletes.

The gender difference appeared more nuanced by observing autonomic indices (Table 5). Nevertheless, as expected [47], males were characterized by an increased RR LFnu, a reduced RR HFnu, an increased arterial pressure (SAP mean) and a slightly reduced Alpha Index as compared to females, suggesting an autonomic nervous system cardiac regulation shifted towards an enhanced sympathetic modulation. No significant difference was noted in ANSIs: this index was built, in fact, in order to correct for gender and age differences, rendering it more suitable to be used to define possible relationships among autonomic nervous system control and other parameters such as, in the present study, data deriving from body composition evaluation and biochemical data.

The two Spearman correlation matrices reported in Figure 1a for females and 1b for males, show the strength of the bivariate relationships among the indices of cardiac autonomic regulation (both considering all the single indices derived from the analysis and considering the unitary ANSIs), arterial pressure, indices of body composition and a relevant set of biochemical indices.

A color-coded presentation highlights that, more evidently in males, autonomic indices (HR, RR Mean, RR Ro, Alpha Index, RR TP, RR LFa, RR HFa, RR LFnu, RR HFnu, RR LF/HF and RR P_0v) showed a significant correlation with body composition indices (in particular, FM%, FFM%, fat mass). ANSIs correlated with % indices of fat and fat free mass, and, in addition, with HDL and glycaemia. In females, a significant correlation between autonomic indices and body composition/metabolic indicators was apparent, particularly for the baroreflex Alpha Index and ANSIs (with FM% and FFM%).

Figure 2 depicts the results of EFA: this approach to data analysis captures more than 70% of information embedded in the data set, as expressed by the total variance. The 23 variables selected from the autonomic modulation analysis, body mass composition and biochemical analysis, together with gender (0 = female, 1 = male), linked to six different common factors (latent domains) of progressively decreasing strength (from 15.03% for factor 1 to 9.81% for factor 6). Of these, the first three factors each explained more than 12% of the total variance and might thus be considered of major importance. The scree plot (upper panel, Figure 2) proved that the remaining factors from number seven onwards were negligible. It may be noted that: the first factor (15.03% VAF) aggregated all dimensionless HRV-derived variables (expression of frequency domain analysis of HRV). The second factor (14.04% VAF) aggregated gender and other parameters that, as expected, are influenced by it (HDL cholesterol, waist circumferences, BMI, and above all, raw values of body mass, in kg). The third factor (12.60% VAF) aggregated body composition data expressed in % and fat mass in Kg. The fourth factor (10.88% VAF) aggregated indices of HRV power expressed in absolute values (expression of time domain analysis of HRV). The fifth factor (10.87% VAF) aggregated all other biochemical parameters, while the sixth factor (9.81% VAF) aggregated the remaining variables.

Finally, Table 6 reports a synthesis of the multiple quantile regression analysis of ANSIs versus the six latent domains as given by the extracted common factors. It should be noted that in addition to a significant relationship, as expected, with latent domains 1, 4 and 6 (all aggregating variables derived from HRV analysis), latent domain 3 (aggregating mainly body composition data expressed in %) was also significantly linked to ANSIs. Latent domains 2 and 5 instead were not significantly linked with ANSIs. Moreover, the latent domains 6 and 1, considered in this order, proved to have the largest effect sizes on ANSIs, given their highest values of standardized beta coefficients (in absolute value) and ∆Pseudo R^2^.

## 4. Discussion

In this retrospective, observational, proof of concept investigation on Olympic athletes, we observed a clear correlation between autonomic nervous cardiac modulation indices and body mass composition parameters, as indicated by both the Spearman correlation matrix computed for single cardiac autonomic variables and variables derived from the analysis of body composition, as well as by the quantile regression result of ANSIs versus the latent domain aggregating mainly body composition data expressed in % (factor 3).

### 4.1. ANSIs as a Proxy of Cardiac Autonomic Regulation in Athletes

The study of cardiac autonomic modulation using a simple, non-invasive methodology, may offer great chances in a clinical setting. Nevertheless, many methodological/interpretative [48] pitfalls in the last decades have hindered this opportunity. In particular, the common use of multiple indicators considered as indices of either vagal or sympathetic cardiac functionality (see e.g., [15]) in the absence of an integrated approach may hamper a reliable inference on underlying neural mechanisms. Consequently, it is often only experts who can appreciate the physiological or pathophysiological role of cardiac neural (dys) regulation either in clinical (where ANS might play a role in multiple important conditions: from sudden coronary death to obesity or cancer) or physiological settings (e.g., in determining athletes’ competition results) [2]. Moreover, we may also consider that in exercise/sport settings, one specific variable (the total variance of RR interval variability, in this paper defined as RRTP), being easy to detect (particularly when using 24-h ECG recordings), caught most of the attention of researchers/clinicians. This variable, considered as a marker of vagal cardiac modulation, may well reflect the effect of aerobic training, while it may not be suitable for unveiling the changes in the autonomic nervous system cardiac control characterizing high intensity training and top performances, conditions that are characterized by a shift towards sympathetic dominance in neural cardiac regulation [2,21]. Keeping all these considerations in mind, we planned to obtain an index of autonomic cardiac regulation in sports (ANSIs) that would be free from gender and age bias, integrating in a single index proxy information on supine rest, standing up and exercise domains [31], and that could be simple to apply in a practical setting (clinical or physiological). Accordingly, the various cardiac autonomic markers were integrated into a unitary proxy by way of a radar plot. Furthermore, the use of a percentile ranking permitted an individual evaluation within the reference population [47]. Summarizing, the unitary index (ANSIs) we adopted attributes a measure of performance (0–100, higher is better) of cardiac autonomic regulation to every athlete. For simplicity, only autoregressive spectral analysis [38] and dynamic evaluation [42]-derived indices were employed, in the context of a simplified cybernetic dual-fed back model [49]. Of particular interest is that ANSIs, which is built considering only variables derived from heart rate variability, correlated well with Alpha Index (a marker of the overall gain of cardiac baroreflex sensitivity) both in male and female athletes. These data are corroborated by previous data of our group [45], which showed a strong correlation between a similar index (ANSI) and cardiac baroreflex sensitivity (Alpha Index) in a relatively large general population. These data suggest that proxy indices estimated from RR variability analysis (such as ANSIs) may be useful to derive information that correlates to cardiac baroreflex sensitivity, which plays an important role in exercise training, with economic and organizational advantages not requiring the need to record blood pressure variability continuously.

### 4.2. Body Composition in Olympic Athletes

Knowledge of body composition is a key element of metabolic assessment, especially in athletes [33], where frequent accurate measures [34] are the key aspects of training assessment. Accordingly, we based our investigation on a simple measuring method (BOD POD) with low intrinsic error (±3%) in order to obtain accurate estimates of body fat categories and body fat distribution for both genders [50] in a large cohort of elite athletes of different sport disciplines. This aspect might be considered a point of critical strength of the present study, taking into account the importance of studying body composition, using state of the art non-invasive methodology, in a large (*n* = 583) population of elite (they all took part in the selection procedure for Olympic games) athletes of both genders. To the best of our knowledge, previously published papers only considered either a small number of athletes or non-elite athletes, or employed less accurate methodologies [33,51,52,53].

### 4.3. Relationship between Body Composition and Autonomic Cardiac Regulation

The link between body composition and autonomic cardiac regulation may be of great practical interest: both these areas are intertwined in determining athletes’ performance and their clinical assessment might be of interest to optimize it. With aerobic training there is a loss of fat and an increase in proxies of vagal cardiac control [54]. A role of the parasympathetic nervous system in weight control is also suggested by recent experimental findings [28], while a multitude of studies, conversely, focused on the simpler relationship of BMI with indices of cardiovascular sympathetic activity [55]. Relatively few studies have addressed the role of body composition [56] showing that a greater share of fat correlates with an increased sympathetic cardiac index. In this study, we had the possibility to investigate, in a large cohort of elite athletes of different sport disciplines, body composition employing the best non-invasive technology. In the present study, we observed a clear correlation between neural autonomic cardiac modulation and body mass composition parameters, as indicated by both the Spearman correlation matrix (Figure 1) and the quantile regression result of ANSIs versus the latent domain 3, which aggregated mainly body composition data expressed in % (Table 6). Many mechanisms may be considered in order to explain the link between ANS and body composition both in the general population and in athletes. In this latter case, we may consider irisin [57] as a link between metabolism and exercise or improved autonomic regulation [58]. On the contrary, ANSIs did not show any significant relationship with factor 2 (which aggregated HDL cholesterol, BMI, waist circumference, body mass and free fat mass expressed in kg) and with factor 5 (which aggregated biochemical parameters). These data may reflect, from one side, the small extension of the biochemistry data range and, from the other side, the health status of this peculiar healthy population. Instead, the observed correlation between indices of autonomic cardiac regulation and indices of body composition (Figure 1) may suggest that a specific training may elicit parallel adaptation either in ANS cardiac control or in body composition, thus underlining the complex multivariate relationship among cardiovascular autonomic regulation and other systems’ components and the need to consider all of them.

### 4.4. Limitations

Some limitations of our study should be acknowledged. First, our findings are derived from Olympic athletes and thus cannot be easily extrapolated to normal untrained individuals. Second, our study is based on a retrospective analysis of data previously collected for other purposes. Their collection, however, was standardized, so that it might be possible to replicate the results of our analysis since the data we have considered were obtained from the clinical procedure employed routinely for the yearly preparticipation assessment of Olympic athletes in Rome. Moreover, in this article we focus only on the relationship between ANS and body composition. Actually, there are other important factors that lead athletes to increase their performance such as, for instance, cardiorespiratory fitness, muscular strength and psychological features, which all interact simultaneously.

## 5. Conclusions

In this investigation on Olympic athletes, we observed a clear correlation between neural autonomic cardiac modulation and body mass composition parameters, suggesting that specific training may elicit parallel adaptation in cardiac ANS modulation and in body composition. This observation, albeit preliminary, may add some knowledge to be potentially considered in order to help athletes to improve their performance close to cardiorespiratory fitness, muscular strength and psychological features. Moreover, this observation can support the growing amount of data showing an important link between the autonomic nervous system and metabolic controls, a link which may be important to better understand the increased risk of chronic non-communicable disease present in overweight/obese subjects.

## Figures and Tables

**Figure 1 jpm-12-01508-f001:**
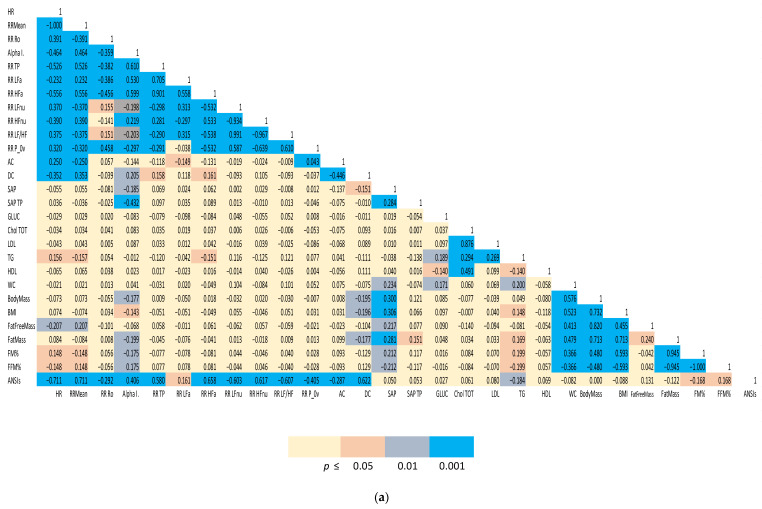
(**a**). Spearman correlation matrix of autonomic, biochemical and body composition in females. (**b**) Spearman correlation matrix of autonomic, biochemical and body composition in males. Color-coded Spearman correlation matrix of selected autonomic and metabolic indices in Olympic athletes, separately computed for females (top panel) and males (bottom panel). Notice that both correlation (r in each cell) and significance level (color-coded cells) are indicated.

**Figure 2 jpm-12-01508-f002:**
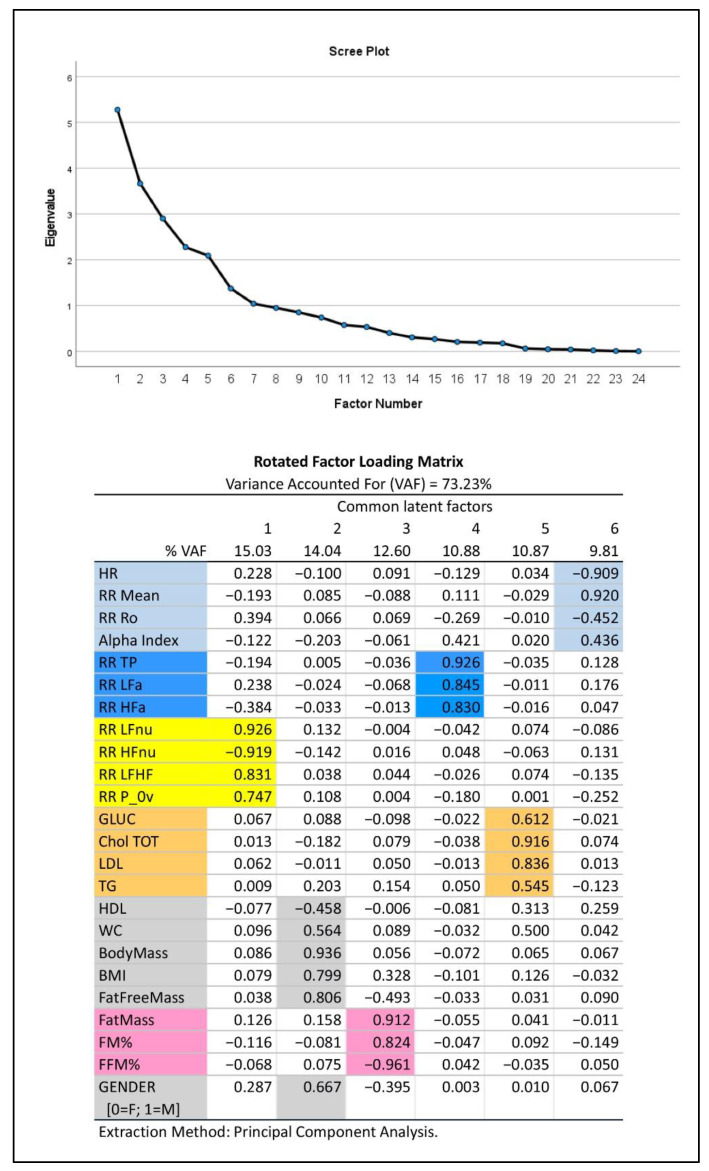
Exploratory Factor Analysis (EFA) of major autonomic and metabolic variables in Olympic athletes. Top panel: Scree plot. Bottom panel: color-coded representation of the rotated factor loadings. Colored cells depict clusters of variables based on their link with the extracted common factors, which represent the primary latent domains to consider. These domains (ordered in decreasing sense according to the individual VAF) are interpreted through the factor loadings as follows: oscillatory (latent domain 1, yellow), body mass and gender (latent domain 2, grey), body composition (latent domain 3, pink), amplitude (latent domain 4, blue), biochemistry (latent domain 5, ocher) and pulse (latent domain 6, light blue).

**Table 1 jpm-12-01508-t001:** List of the abbreviations and definitions of variables employed in the study.

Variables	Units	Definition
Alpha Index	ms/mmHg	Alpha index is a frequency domain index of cardiac baroreflex
BRS	ms/mmHg	Cardiac baroreflex sensitivity
HR	beat/min	Heart rate, i.e., number of cardiac beats per minute
RR mean	ms	RR interval obtained from a single lead ECG tachogram
RR TP	ms^2^	RR total variance obtained from a single lead ECG tachogram
RR LFa	ms^2^	Absolute power (a) of low frequency (LF) autoregressive spectral component of RR variability (V)
RR HFa	ms^2^	Absolute power (a) of high frequency (HF) spectral component of RRV
RR LFnu	nu	Normalized power (nu) of low frequency (LF) spectral component of RRV
RR HFnu	nu	Normalized power (nu) of high frequency (HF) spectral component of RRV
RR LF/HF	.	Numerical ratio between LF and HF spectral powers from tachogram
ΔLFnu	.	Difference in LF power in nu between stand and rest
RR Ro	.	Tachogram index of self-similarity (range 0–1)
RR P_0v	.	Frequency of tachogram three beat pattern, no variations
RR P_2uv	.	Frequency of tachogram three beat pattern, unlike variations
AC	msc	Acceleration capacity index of the tachogram
DC	ms	Deceleration capacity index of the tachogram
SAP	mmHg	Systolic arterial pressure (resting)
DAP	mmHg	Diastolic arterial pressure (resting)
SAP Mean	mmHg	Mean values of continuous systolic arterial pressure recordings during rest
SAP TP	mmHg^2^	Systolic arterial pressure variance (resting)
SAP LFa	mmHg^2^	Absolute power (a) of low frequency (LF) autoregressive spectral component of SAP variability
SAP HFa	mmHg^2^	Absolute power (a) of high frequency (LF) autoregressive spectral component of SAP variability
BMI	Kg/m^2^	Body mass index
WC	cm	Waist circumference
BSA	m^2^	Body surface area
FM%	%	Fat mass percent (BOD POD)
FFM%	%	Fat free mass percent (BOD POD)
Fat Mass	kg	Measured fat mass (BOD POD)
FatFreeMass	kg	Measured fat free mass (BOD POD)
Body Mass	kg	Measured body mass (BOD POD)
Body Volume	L	Volume of overall body
Chol TOT	mg/dL	Fasting total cholesterol (plasma)
HDL	mg/dL	Fasting high-density cholesterol (plasma)
LDL	mg/dL	Fasting low-density cholesterol (plasma)
TG	mg/dL	Fasting triglyceride (plasma)
GLUC	mg/dL	Fasting glucose (plasma)
Stress perception	.	Perception of stress
Fatigue perception	.	Perception of fatigue
4SQ	.	Subjective stress-related somatic symptoms questionnaire
ANSIs	%	Percentile ranked Autonomic Nervous System Index for sports

**Table 2 jpm-12-01508-t002:** Anthropometric data: summary statistics (median ± MAD) and Mann–Whitney test.

Variables	Females	Males	All	Sig (2-Tailed)
*n*	230	353	583	(F vs. M)
Age (years)	23.00 ± 4.00	24.00 ± 4.00	23.00 ± 4.00	0.842
Weight (kg)	62.00 ± 6.00	79.00 ± 7.00	72.00 ± 9.00	<0.001
Height (m)	1.70 ± 0.05	1.83 ± 0.07	1.78 ± 0.08	<0.001
SAP (mmHg)	111.00 ± 6.00	125.00 ± 7.00	119.00 ± 9.00	<0.001
DAP (mmHg)	62.00 ± 4.00	65.00 ± 5.00	64.00 ± 4.00	<0.001
WC (cm)	75.00 ± 5.00	84.00 ± 4.00	81.00 ± 5.50	<0.001
BMI (kg/m^2^)	21.30 ± 1.23	23.29 ± 1.49	22.63 ± 1.80	<0.001

Abbreviations: Sig = MW test significance level; SAP = systolic arterial pressure; D = diastolic; WC = waist circumference; BMI = body mass index.

**Table 3 jpm-12-01508-t003:** Common indicators of body composition: summary statistics (median ± MAD) and Mann–Whitney test.

Variables	Females	Males	All	Sig (2-Tailed)
FM% (%)	19.35 ± 3.55	11.10 ± 3.00	13.50 ± 4.90	<0.001
FFM% (%)	80.65 ± 3.55	88.80 ± 3.00	86.40 ± 5.00	<0.001
Fat Mass (kg)	11.91 ± 3.10	8.65 ± 2.94	10.05 ± 3.35	<0.001
FatFreeMass (kg)	50.51 ± 4.47	69.43 ± 6.90	61.80 ± 9.67	<0.001
Body Mass (kg)	62.57 ± 6.08	78.46 ± 7.77	72.60 ± 9.30	<0.001
Body Volume (L)	59.13 ± 5.80	73.40 ± 7.35	68.44 ± 8.72	<0.001
BSA (m^2^)	1.72 ± 0.11	2.00 ± 0.13	1.89 ± 0.16	<0.001

Abbreviations: Sig = MW test significance level; BSA = body surface area.

**Table 4 jpm-12-01508-t004:** Major metabolic and subjective stress indicators: summary statistics (median ± MAD) and Mann–Whitney test.

Variables	Females	Males	All	Sig (2-Tailed)
Chol TOT (mg/dL)	182 ± 22	168 ± 20	173 ± 22	<0.001
HDL (mg/dL)	74 ± 10	61 ± 9	66 ± 10	<0.001
LDL (mg/dL)	93 ± 18	91 ± 18	92 ± 18	0.989
TG (mg/dL)	64 ± 13	64 ± 17	64 ± 15	0.624
GLUC (mg/dL)	92 ± 4	95 ± 4	94 ± 4	<0.001
Stress perception (.)	2 ± 2	2 ± 2	2 ± 2	0.120
Fatigue perception (.)	3 ± 2	3 ± 2	3 ± 2	0.387
4SQ (.)	17 ± 14	12 ± 11	14 ± 12	0.006

Abbreviations: Sig = MW test significance level; Chol TOT = total cholesterol; HDL = high-density lipoprotein; LDL = low-density lipoprotein; TG = triglyceride; GLUC = glucose; 4SQ = subjective stress-related somatic symptoms questionnaire.

**Table 5 jpm-12-01508-t005:** Cardiac autonomic indices: summary statistics (median ± MAD) and Mann–Whitney test.

Variables	Females	Males	All	Sig (2-Tailed)
HR (bpm)	59.70 ± 7.23	57.73 ± 6.37	58.28 ± 6.57	0.029
RR mean (ms)	1005.02 ± 120.85	1039.38 ± 112.94	1029.57 ± 116.14	0.029
RR TP (ms^2^)	3672.69 ± 1902.73	3356.49 ± 1610.03	3493.89 ± 1716.71	0.374
RR LFa (ms^2^)	638.86 ± 339.92	918.18 ± 505.07	784.14 ± 446.78	<0.001
RR HFa (ms^2^)	1484.47 ± 1008.77	918.61 ± 615.45	1120.38 ± 761.38	<0.001
RR LFnu (nu)	30.94 ± 12.62	42.86 ± 15.20	38.43 ± 14.59	<0.001
RR HFnu (nu)	62.94 ± 14.45	50.75 ± 15.39	56.24 ± 15.37	<0.001
RR LF/HF	0.48 ± 0.28	0.86 ± 0.49	0.68 ± 0.41	<0.001
Alpha Index (ms/mmHg)	27.80 ± 8.93	22.80 ± 7.98	24.80 ± 8.51	<0.001
BRS (ms/mmHg)	24.70 ± 9.47	21.16 ± 8.82	22.15 ± 9.03	0.007
RR Ro (.)	0.22 ± 0.06	0.23 ± 0.07	0.22 ± 0.06	0.076
SAP Mean (mmHg)	110.84 ± 10.12	125.58 ± 9.79	118.86 ± 11.02	<0.001
SAP TP (mmHg^2^)	13.12 ± 7.64	24.64 ± 14.16	19.97 ± 11.96	<0.001
SAP LFa (mmHg^2^)	1.81 ± 1.13	2.82 ± 1.79	2.33 ± 1.48	<0.001
SAP HFa (mmHg^2^)	0.92 ± 0.50	1.14 ± 0.68	1.03 ± 0.59	0.086
Δ LF (nu)	45.47 ± 13.28	39.09 ± 15.60	42.65 ± 14.51	0.002
ANSIs (%)	49.81 ± 25.95	50.58 ± 24.04	50.28 ± 25.00	0.692
RRP_0v (.)	8.80 ± 5.22	14.00 ± 7.50	11.50 ± 6.34	<0.001
RRP_2uv (.)	31.62 ± 10.51	26.47 ± 9.98	28.38 ± 10.15	<0.001
AC (ms)	2.95 ± 0.52	3.12 ± 0.59	3.05 ± 0.57	0.006
DC (ms)	3.47 ± 0.93	2.96 ± 0.96	3.16 ± 0.98	0.032

Abbreviations: Sig = MW test significance level; HR = heart rate; TP = total power, i.e., variance; LF = low frequency; a = absolute value; nu = normalized units; HF = high frequency; BRS = baroreflex sensitivity; Ro = regularity index; SAP = systolic arterial pressure; ANSIs = Autonomic Nervous System Index for sport; RRP = symbolic pattern; AC = acceleration capacity; DC = deceleration capacity.

**Table 6 jpm-12-01508-t006:** Multiple quantile regression model of ANSIs against biochemical, body composition and autonomic latent domains in Olympic athletes: estimation and significance test for the latent domain effect sizes.

Latent Domain	Standardized B Coefficient ^(^*^)^	t ^(^**^)^	Significance ^(^**^)^	∆Pseudo R^2 (^***^)^
1	−0.545	−13.170	**<0.001**	0.171
2	0.064	1.533	0.126	0.004
3	−0.130	−3.080	**0.002**	0.012
4	0.254	6.251	**<0.001**	0.049
5	0.029	0.584	0.559	0.001
6	0.615	14.247	**<0.001**	0.204
Model quality measures: Pseudo R^2^ = 0.397; Mean Absolute Error (MAE) = 14.883

Note: The meaning of the six latent domains (represented by the extracted common factors) is provided below Figure 2. ^(^*^)^ Standardized beta coefficients β^j* are computed with the general formula: β^j*=β^jσ^jσ^y, where β^j is the coefficient estimate for factor j in the multiple quantile regression, σ^j is the standard deviation of factor j, which is equal to 1 by construction, and σ^y=28.7423 is the computed standard deviation of ANSIs. ^(^**^)^ Student *t*-test is based on the asymptotic normality of the parameter estimators. Significant regression coefficients are in bold. ^(^***^)^ The ΔPseudo R^2^ index is defined as: ΔPseudo R^2^ = Pseudo R^2^ (mod_all_)—Pseudo R^2^ (mod_-j_), where Pseudo R^2^ (mod_all_) refers to the model containing all the six latent factors, while Pseudo R^2^ (mod_-j_) refers to the model that excludes factor j from the independent variable set, j = 1,…, 6.

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
