# Peer review of "Relationship between Body Composition and Cardiac Autonomic Regulation in a Large Population of Italian Olympic Athletes"

_jpm, 2022, doi:10.3390/jpm12091508_

Round 1

Reviewer 1 Report

This is a well written article on the association between body composition and cardiac autonomic regulation among athletes.  I have some comments to further strengthen the analysis description for readers.

1) there is a mix of using parametric (ie, t-test, EFA) and non-parametric (ie, spearman correlation coefficient) statistical methods.  It is necessary to present distributional assessment in order to run parametric tests.  Please include or use non-parametric methods throughout.  

2) Please provide more detail on how the number of latent factors were selected?

3) Figure 2 - overall there are clear separation of the latent factors as evident by high factor loadings.  What criteria did you use to only select a single latent factor for each variable in the EFA?  Some variables appear to load on more than 1 latent factor.  For example, Alpha index loads on both latent factor #6 and #4. 

4) The terms "component", "latent domain", and "latent factor" are used interchangeably.  Suggest using "latent factor" or "latent domain" throughout.

5) Table 6 - are the latent factors all used in a single linear regression model or are they separate models?  Are any other variables in the models?

6) where there any differences in the factor loadings to suggest running the EFA stratified by sex?

Author Response

see uploaded pdf version

Reviewer 2 Report

This is an interesting work based on a large sample of Italian Olympic athletes, where heart rate variability (HRV) indices were correlated with body composition variables "...to verify the strength of the interplay among a number of major elements of training, including autonomic nervous system (ANS) modulation, biochemical indicators and body composition, in a system medicine approach." The study motivation in the Introduction section is clear and appealing. The methods used to measure such variables are adequate, and the results are interesting. However, there are some issues that should be addressed, as described below.

(1) The study included Italian Olympic athletes only. The sample is large but does not necessarily represent phenotypic differences of athletes from other regions of the world. I strongly recommend adding the word "Italian" to the title. 

(2) The methods used to measure the study variables are adequate, but some technical details are needed. The software used for HRV was previously described, as mentioned by the authors (lines 149-150). However, a brief description of the parameters is necessary to allow reproducibility of the methods in future works by other authors. For instance, for HRV analysis, it is important to mention the steps for the spectral analysis (identification and treatment of RR intervals from arrhythmias, detrending, windowing, and parameters for the autoregressive model).

(3) The statistical analysis should be revised. Were the variables tested for normal distribution? Please clarify how the authors decided when to use parametric or non-parametric statistical methods. It is confusing why the analysis consisted of Spearman correlation (non-parametric) and multivariate linear regression analysis (parametric).

(4) The results presentation and inferential tests should be improved according to the variables' distribution using parametric or non-parametric statistics. Results are shown as mean and standard deviation, but based on the results, several variables clearly do not follow a normal distribution (for instance, from Table 5, RRTP, RRHFa, BRS, SAPTP).

(5) The Conclusions section should improve, addressing with more detail the actual knowledge acquired with the results of the present work and in connection with the study aim ("...to verify the strength of the interplay among a number of major elements of training, including autonomic nervous system (ANS) modulation, biochemical indicators and body composition, in a system medicine approach"). The potential relationships of the studied variables with "specific training", "sport performance" and "winning phenotype" were not assessed in the present work. The ideas mentioning these important sports variables are better placed in the Discussion section, where the potential applications can be proposed and based on other works as a guide for the reader.

Author Response

see uploaded pdf version

Round 2

Reviewer 2 Report

The revised manuscript addressed almost all issues. The statistical analysis improved significantly, with the authors choosing a non-parametric approach. However, the results in Tables 3, 4, and 5 should be described by median and interquartile range, or median (percentile 25 - percentile 75), since the Mann-Whitney test compares the median between groups, not the mean values. Describing the results by mean and standard deviation is not appropriate.

Author Response

see uploaded file
